# The Progress and Pitfalls of Pharmacogenetics-Based Precision Medicine in Schizophrenia Spectrum Disorders: A Systematic Review and Meta-Analysis

**DOI:** 10.3390/jpm13030471

**Published:** 2023-03-04

**Authors:** Yuxin Teng, Amrit Sandhu, Edith J. Liemburg, Elnaz Naderi, Behrooz Z. Alizadeh

**Affiliations:** 1Department of Epidemiology, University Medical Centre Groningen, University of Groningen, 9713 GZ Groningen, The Netherlands; 2Department of Psychiatry, Rob Giel Research Center, University Medical Center Groningen, University of Groningen, 9713 GZ Groningen, The Netherlands; 3Center for Statistical Genetics, Gertrude H. Sergievsky Center, and the Department of Neurology, Columbia University Medical Center, New York, NY 10032, USA

**Keywords:** schizophrenia, pharmacogenetics, antipsychotics, personalized psychiatry, CYP, drug response, adverse events, pharmacokinetics, weigh-gain, tardive dyskinesia

## Abstract

The inadequate efficacy and adverse effects of antipsychotics severely affect the recovery of patients with schizophrenia spectrum disorders (SSD). We report the evidence for associations between pharmacogenetic (PGx) variants and antipsychotics outcomes, including antipsychotic response, antipsychotic-induced weight/BMI gain, metabolic syndrome, antipsychotic-related prolactin levels, antipsychotic-induced tardive dyskinesia (TD), clozapine-induced agranulocytosis (CLA), and drug concentration level (pharmacokinetics) in SSD patients. Through an in-depth systematic search in 2010–2022, we identified 501 records. We included 29 meta-analyses constituting pooled data from 298 original studies over 69 PGx variants across 39 genes, 4 metabolizing phenotypes of *CYP2D9*, and 3 of *CYP2C19*. We observed weak unadjusted nominal significant (*p* < 0.05) additive effects of PGx variants of *DRD1*, *DRD2*, *DRD3*, *HTR1A*, *HTR2A*, *HTR3A*, and *COMT* (10 variants) on antipsychotic response; *DRD2*, *HTR2C*, *BDNF*, *ADRA2A*, *ADRB3*, *GNB3*, *INSIG2*, *LEP*, *MC4R*, and *SNAP25* (14 variants) on weight gain; *HTR2C* (one variant) on metabolic syndrome; *DRD2* (one variant) on prolactin levels; *COMT* and *BDNF* (two variants) on TD; HLA-DRB1 (one variant) on CLA; *CYP2D6* (four phenotypes) and *CYP2C19* (two phenotypes) on antipsychotics plasma levels. In the future, well-designed longitudinal naturalistic multi-center PGx studies are needed to validate the effectiveness of PGx variants in antipsychotic outcomes before establishing any reproducible PGx passport in clinical practice.

## 1. Introduction

Schizophrenia spectrum disorders (SSD) are severe and chronic neurodevelopmental psychiatric disorders that inversely affect patients’ quality of life. Affected persons can experience auditory and visual hallucinations, detachment from reality, and a distorted ability to think and talk [1]. The lifetime prevalence of SSD worldwide is ~3%, yielding 12.66 (9.48 to 15.56) million disability-adjusted life years (DALYs) in 2017 [2,3]. These disorders are highly heterogeneous among individuals as different key brain systems are differently disturbed during disease development and progression, resulting in differences in patients‘ memory, thinking, decision-making, and or emotional processing [4]. Moreover, SSD affects multiple molecular signaling pathways and their interactions, including dopaminergic, glutamatergic, GABAergic, serotoninergic, and ErbB signaling pathways [5,6].

Despite recent advances in the treatment of SSD, the risk of relapse ranges from 41% to 79% within one year after the first episode, particularly after discontinuing medications [7]. Up to 35% of patients do not respond to antipsychotics; these are treatment-resistant or non-responders [8]. Furthermore, adverse effects of antipsychotics are common and often difficult to manage. This leads to treatment non-compliance and discontinuation and, thus, to an even higher risk of recurrence and deterioration in patients [9]. Suboptimal efficacy and often existing adverse effects impose persistent mental and financial burdens on patients, families, and caregivers, requiring a more beneficial treatment strategy.

Precision psychiatry is an emerging field that strives to create personalized therapeutic strategies for individuals based on their unique health conditions. By leveraging scientific and technological advancements, it aims to deliver timely and effective treatment to each patient. The implementation of precision psychiatry has shown considerable promise, with examples such as the I-SHARED project in the Netherlands, which supports treatment choices for depression using combined socio-demographic and clinical data [10]. Another example is the MOPHAR program, which promptly detects and treats adverse psychotropic medication effects and monitors medication reconciliation [11]. It is recommended that patients taking specific psychotropic medications, such as lithium, carbamazepine, and atypical and typical antipsychotics, undergo regular monitoring of various parameters, such as blood sugar, lipid levels, weight, vital signs, and involuntary movement disorders [12]. Currently, the most idealized approach involves integrating a variety of data sources, including symptoms, physiology, behavior, demographics, lab results, genetics, imaging, cognition, wearable sensors, and life experiences, to accurately diagnose and classify patients according to their clinical biotypes, ultimately leading to the development of targeted treatments [13,14,15]. However, this approach can only be fully realized when each aspect of the data source has a high degree of credibility and reliability.

To establish trustworthy genetic evidence for personalized treatment of SSD, we conducted a systematic review and meta-analysis of peer-reviewed meta-analyses published since 2010 that examined the associations between antipsychotic pharmacogenetics (or pharmacogenomics; PGx) variants and seven treatment outcomes in SSD. The outputs of this study can be used to construct polygenic risk scores (PRSs), combining multiple PGx variants. This could potentially accelerate the implementation of personalized treatment for SSD.

### 1.1. Pharmacogenetics

PGx is a promising method that may aid in improving the efficacy of drugs used to treat diseases such as cancer and hypertension, as well as psychotropics in psychiatric disorders [16,17,18]. PGx aims at identifying the genetic variants that regulate drug response. Therefore, identifying patients based on their genetic predisposition to respond to treatments may lead to more informed drug selection, optimizing therapeutic efficacy while minimizing the risk of adverse events. The U.S. Food & Drug Administration (FDA) has approved the PGx labeling of some antipsychotics [19]. For example, Cytochrome P450 2D6 (*CYP2D6*) is an enzyme catalyzing the metabolism of 40% of antipsychotics [20]. The *CYP2D6* gene is highly polymorphic, and the presence of each variant leads to the production of slightly different enzymes with either increased, normal, decreased, or even absent activities. Carriers of *CYP2D6* non-functioning variant(s) metabolize the corresponding medications more slowly, known as poor metabolizers [21]. Therefore, the FDA recommends *CYP2D6* poor metabolizers should take a reduced dose of aripiprazole and clozapine due to slower metabolism to avoid potentially increasing drug exposure, thus leading to drug complications [22,23]. A review conducted by van Westrhenen et al. concluded that the most clinically relevant genetic variants are associated with the pharmacokinetics of the drug, particularly the variants in *CYP2C19* and *CYP2D6* [24]. Likewise, Murphy et al. reported that the consensus among pharmacogenetic experts is now that *CYP2D6* and *CYP2C19* genotype information enables treatment recommendations for a wide range of antipsychotics [19]. Both studies found a lack of clinical evidence about the role of other various genetic variants, including those at antipsychotic-specific action sites (e.g., *DRD2*), in the treatment outcomes of antipsychotics. Hence, the identification of more genetic variants with potential clinical impact on the outcome of antipsychotic treatment is essential.

### 1.2. Role of Pharmacogenetics in Precision Psychiatry

Precision psychiatry relies on various sources of information to personalize treatment for patients, and one such important source is an individual’s genetic profile. A genetic profile can be easily determined and provides a stable source of information throughout a person’s life. A retrospective study in a psychiatric setting reported that variations in CYP450 activity might be the cause of adverse drug reactions, abnormal therapeutic drug monitoring, or non-response in 34.7% of patients with an intermediate to high probability [25]. Additionally, it is worth noting that around 5–10% of the European population are *CYP2D6* poor metabolizers due to inheriting two nonfunctional alleles [26]. This can significantly affect their response to medications and may require alternative dosing strategies or alternative medications altogether. Therefore, understanding the effect of genetic profiles on treatment outcomes is crucial for advancing precision psychiatry.

### 1.3. PGx Studies Shortcomings

The number of existing PGx studies and subsequent systematic reviews has grown exponentially in the last decade, thanks to the rapid evolution of genetic testing technologies. Although overwhelmingly increasing studies offer more evidence to support the PGx-based decision-making for drug choices, clinicians may be confused by the uncertain validity of the results and contradicting evidence. To elaborate, shortcomings in the existing PGx research, including irreproducible results [27,28], non-standardized measurements [29,30], limited samples [31], the discrepancy in results reporting [32], as well as imprecise use of genetic terminologies [33], can significantly affect PGx clinical applicability. Moreover, most existing reviews either focused on only one specific outcome such as drug efficacy, antipsychotic-induced weight gain, and only one gene [34,35], or just offered an overview without providing detailed quantitative evidence supporting their recommendations [36,37]. These facts make it hard for clinicians to make an optimal drug choice. As a result, beneficial decisions about drug choices could be fundamentally hindered, leading to an ineffective translation from PGx research findings to improving treatment efficacy in SSD clinical practice [19]. The questions raised are, what is the current state of PGx research in SSD, and what is the quality of evidence that has been implemented as PGx guidelines offered for routine practice?

## 2. Materials and Methods

The latest Preferred Reporting Items for Systematic Reviews and Meta-Analyses (PRISMA) guidelines were followed in this study [38].

### 2.1. Search Strategy and Study Selection

A search strategy using the PICO format was developed. The original search was conducted in January 2022 and updated in October 2022. MEDLINE, Embase, CINAHL, PsycINFO, and Web of Science were searched using keywords and index terms such as: (1) terms for population: ‘schizophrenia’ OR ‘antipsychotic’ OR ‘neuroleptic’; (2) terms for intervention: ‘pharmacogenetic’ OR ‘genome wide association’; (3) terms for outcome: ‘adverse reaction’ OR ‘treatment outcome’ OR ‘efficacy’; (4) terms for study design: ‘meta-analysis’.

The search was limited to include only articles written in English and published in the last 12 years, as the study design and genetic techniques used in PGx have been modernized and remain comparable across the original studies and systematic reviews. Articles were considered eligible if they (1) were meta-analyses with a systematic search strategy for previously published original studies, (2) included people with a diagnosis of SSD or psychosis, and (3) investigated any association between genetic variants and treatment outcomes of antipsychotics. All articles were screened first by viewing the title and abstract level. Articles that met the inclusion criteria were reviewed at the full-text level.

### 2.2. Outcomes

We included treatment responses to antipsychotics, antipsychotic-induced weight gain, BMI, metabolic syndrome, tardive dyskinesia (TD), clozapine-induced agranulocytosis (CIA), antipsychotic-related prolactin levels, and plasma levels of antipsychotic drugs.

### 2.3. Data Extraction and Synthesis

Data extraction was performed independently by Y.T. and A.S. Any discrepancies that arose were discussed, and decisions were made by agreement among evaluators. For each eligible meta-analysis study, data were extracted, including PMID/doi, the name of the first author, year of publication, review coverage, original research studies included in the meta-analysis, antipsychotic type, outcome/endpoints, the model used for meta-analyses, number of original studies included, number of pooled sample size, study population, genes, single nucleic polymorphisms (SNPs), comparisons, pooled effect size metrics used in the meta-analyses with 95% confidence intervals (CIs), and heterogeneity between included original studies.

### 2.4. Data Synthesis and Analysis

The coverage of included meta-analysis studies was visualized using a network diagram created with Flourish (https://flourish.studio/, accessed on 3 November 2022). This revealed and visualized the distribution and the overlap of original studies across different meta-analysis studies, offering a comprehensive overview of the scope of the present study.

Meta-analyses summary statistics, including genetic models, odds ratio, standard errors, confidence intervals, p values, heterogeneity tests, and the number of samples for each reported PGx variant, were extracted and presented in the supplementary tables. These data were ensembled per order of studied outcome, population, study ID, gene name, and variants RSnumber. Genetic models included dominant, recessive, and additive models. The measure of associations was reported as odds ratio, Hedges’g, mean difference, or ratio of means.

If multiple meta-analyses reported the same PGx variant and outcome using the same effect size metric, a new meta-analysis was conducted by combining all the original studies to recalculate the effect sizes. We used MetaGenyo (http://bioinfo.genyo.es/metagenyo, accessed on 10 November 2022), a web tool for meta-analysis of genetic association, to estimate meta-effect sizes [39]. When raw data were not available to perform a new meta-analysis, the results from the meta-analysis study with the largest population size were reported. We performed sensitivity analyses with stratifications by ethnicity where possible.

### 2.5. Additive Effects

We identified SNPs which had significant reports in any of the genetic models for any of the desired outcomes and presented the additive effect, assuming a codominant effect of alleles when no confirmed data was available to support a dominant or recessive effect. The quantitative additive data were derived from either the additive results reported in the meta-analysis studies or our own calculations using MetaGenyo when the results were not reported under the additive model.

## 3. Results

### 3.1. Study Inclusion

The systematic search yielded a total of 501 records. After the removal of 80 duplicates, 421 titles and abstracts were examined, and 103 potentially relevant full texts were retrieved. We excluded 119 articles which did not study SSD, 187 which did not examine the association between PGx variants and treatment outcomes, and 12 which were published before 2010 (Figure 1). We successfully retrieved 97 full texts, and 28 eligible meta-analysis studies were identified after excluding those not in English (*n* = 4), conference abstracts (*n* = 10), and articles without a meta-analysis (*n* = 39) or without a systematic search (*n* = 16), and one additional study was identified and included through manual search of references, as presented in the PRISMA flowchart (Figure 1).

The 29 included meta-analysis studies pooled data from 298 original studies, covering 69 PGx variants, 4 phenotypes of *CYP2D9*, and 3 phenotypes of *CYP2C19*, mapped in 39 genes. They meta-analyzed the PGx variants in association with an antipsychotic response (11 meta-analysis studies; 20 variants across 10 genes), antipsychotic-induced weight/BMI gain (eight meta-analysis studies; 38 variants across 20 genes and four phenotypes of *CYP2D6*), metabolic syndrome (one meta-analysis study; three variants in *HTR2C*), antipsychotic-related prolactin levels (two meta-analysis studies; two variants in *DRD2* and four phenotypes of *CYP2D6*), antipsychotic-induced tardive dyskinesia (four meta-analysis studies; five variants across four genes), clozapine-induced agranulocytosis (one meta-analysis study; 13 PGx variants across nine genes), and plasma levels of antipsychotic drugs (three meta-analysis studies; two PGx variants in *CYP1A2*, four phenotypes of *CYP2D6* and three phenotypes of *CYP2C19*) (Appendix A). For treatment response, three meta-analysis studies reported in a mixed population, Caucasians and Asians separately [40,41,42]; one meta-analysis study reported in a mixed population and Caucasians [31]; six meta-analysis studies reported in mixed populations only [34,43,44,45,46,47]; one meta-analysis study reported in Asians [48]. For antipsychotic-induced weight/BMI gain, two meta-analysis studies reported in a mixed population, Caucasians and Asians separately [49,50]; two meta-analysis studies reported in a mixed population and Asians separately [51,52]; three meta-analysis studies reported in mixed population only [35,53,54]; one meta-analysis study did not state the ethnicity of the population [32]. One meta-analysis study reported antipsychotic-induced metabolic syndrome in a mixed population, Caucasians and Asians separately [49]. For antipsychotic-related prolactin levels, one meta-analysis study reported in a mixed population, Caucasians and Asians separately [55], while the other one did not state the ethnicity [56]. For antipsychotic-induced tardive dyskinesia, two meta-analysis studies reported in a mixed population, Caucasians and Asians separately [57,58], and the other two reported in mixed populations only [59,60]. One meta-analysis study reported clozapine-induced agranulocytosis in mixed populations [61]. For antipsychotic exposure (pharmacokinetics), one meta-analysis study reported in a mixed population and Asians [62], while the other two reported in a mixed population only [63,64].

### 3.2. Original Research Studies Included in Meta-Analysis Studies

As shown in Figure 2, one original study can be included in several meta-analysis studies, resulting in a data overlap. For example, each of the original studies included in Study 9 (Yoshikawa, 2020 [45]) was also included in Study 7 (Takekita, 2016 [44]). Original studies investigating more than one outcome were included in meta-analyses on different outcomes. For instance, the original study by Zai et al. [65] investigated both antipsychotic response and AIWG; thus, it was included in both Study 6 (Cargnin, 2016 [40]) and Study 15 (Zhang, 2016 [32]). The same rule applies to other studies (Figure 2). The figure and complete details can be found at Weblink 1.

### 3.3. Genes Investigated in Included Meta-Analysis Studies

We presented all the analyzed genes per type of medication and outcome in each study in Figure 3. The number of PGx variants analyzed for each gene, and the number of PGx variants with significant reports were stated. For instance, Gressier et al. (Study 5) investigated the association of three *HTR2A* PGx variants (rs6313, rs6311, and rs6314) with clozapine response, and they found two PGx variants (rs6313 and rs6314) exhibited significant results [31]. As for *CYP2D6* and *CYP2C19*, the number of phenotypes of the corresponding enzyme was reported. For example, the study by Calafato et al. analyzed four metabolizer statuses of *CYP2D6*, including poor metabolizers (PMs), intermediate metabolizers (IMs), extensive metabolizers (EMs), and ultrarapid metabolizers (UMs) [56] (Figure 3).

### 3.4. Additive Effect Sizes

In Figure 4, we displayed the quantitative additive effect sizes of the PGx variants with significant associations under any genetic models from included meta-analyses or the results we recalculated by combining two or more meta-analyses. Study ID 0 represents the present study. The association of *HTR2A* rs6313 with clozapine response in mixed populations and Asians was reanalyzed in the present study. Several PGx variants showed significant associations only under other genetic models than an additive model, such as *HTR2A* rs6311 with olanzapine response and *COMT* rs4680 with TD in women.

### 3.5. Antipsychotics Response

Overall, 11 out of 29 included meta-analysis studies investigated the antipsychotic response. The studied antipsychotics included clozapine (five meta-analysis studies [31,34,40,41,47]), risperidone (two meta-analysis studies [41,48]), olanzapine (one meta-analysis study [41]), atypical antipsychotics combined together (two meta-analysis studies [40,41]), and all antipsychotics combined together (seven meta-analysis studies [40,42,43,44,45,46,47]). These studies examined the associations between 10 genes and response to either a specific antipsychotic or multiple antipsychotics combined (Figure 3, Appendix A). The response was evaluated using different methods, including the Global Assessment Scale (GAS), personal interview, Brief Psychiatric Rating Scale (BPRS), Basal Positive Symptom Subscale (BPOS), Basal Negative Symptom Subscale (BNEG), Positive and Negative Syndrome Scale (PANSS), and Clinical Global Impression of Improvement (CGI-I). The studied genes included *DRD1*, *DRD2*, *DRD3*, *HTR1A*, *HTR2A*, *HTR2C*, *HTR3A*, *TNFα*, *COMT*, and *BDNF*. At least one PGx variant in each of the seven genes (*DRD1*, *DRD2*, *DRD3*, *HTR1A*, *HTR2A*, *HTR3A*, and *COMT*) exhibited statistically significant associations under at least one of the three genetic models, i.e., dominant, recessive, or additive. Other analyzed PGx variants in the remaining four genes (*DRD3*, *HTR2C*, *TNFα*, and *BDNF*) showed no significant associations (Figure 3, Appendix A).

#### 3.5.1. *HTR2A*

We pooled data from two meta-analysis studies (Gressier, 2016 [31] and Yan, 2022 [41]) investigating the association between *HTR2A* rs6313 and clozapine response. We found a significant additive effect only in Caucasians, with the T allele being associated with a higher chance of response to clozapine (pooled OR: 1.35, 95% CI: 1.01, 1.81; four original studies). The association was not significant in Asians, where the C allele was the effect allele (pooled OR: 1.23, 95% CI: 0.86, 1.76; three original studies). The two other PGx SNPs, the *HTR2A* rs6314*C and *HTR3A* rs1062613*T, exhibited significant associations with response to clozapine, with a pooled OR of 1.75 (95% CI: 1.20, 2.56; five original studies) and 2.13 (95%CI: 1.08, 4.17; four original studies), respectively [31] (Figure 4, Appendix A). Yan et al. reported that the *HTR2A* rs6311*AA genotype was significantly associated with a better olanzapine response compared to G carriers, with a pooled OR of 1.85 (95%CI: 1.18, 2.90; two original studies). However, the additive model (i.e., the A allele compared to the G allele) showed no significant result (Figure 4, Appendix A) [41].

#### 3.5.2. *DRD1*, *DRD2* and *DRD3*

Patients with the *DRD1* rs11746641*GG genotype compared to TT and SerCys in *DRD2* rs1801028 compared to SerSer were found to benefit more from using risperidone with a pooled weighted mean difference of 4.92 (95% CI: 1.21, 8.63; one original study) and 11.58 (95% CI: 5.81, 17.35; two original studies) in PANSS total score in a Chinese population [48]. Yet, no additive models were analyzed or raw data provided. While Hwang et al. did not find any association between *DRD3* rs6280 and clozapine response in mixed populations of Caucasians and African Americans [66], Liu et al. reported that the rs6280*C allele was significantly associated with better treatment response to antipsychotics in Caucasians with a pooled odds ratio of 1.39 (95% CI: 1.12, 1.72; seven original studies), but not in Asians or African Americans [42].

#### 3.5.3. *COMT*

The carriers of the Methionine (Met) allele of the *COMT* rs4680 responded better to atypical antipsychotics compared to the Valine (Val) allele (pooled OR: 1.2, 95% CI: 1.01, 1.44; 15 original studies) [43]. This association did not remain significant with typical antipsychotics, despite the larger sample size (pooled OR: 1.13, 95% CI: 0.96, 1.34; 15 original studies) [43].

#### 3.5.4. *HTR1A*

There were two meta-analysis studies (Takekita, 2016 [44] and Yoshikawa, 2020 [45]) looking into *HTR1A*. The larger study on the *HTR1A* rs6295 showed that the rs6295*C allele was associated with improvement in negative symptoms in an additive model (pooled SMD: 0.14, 95% CI: 0.01, 0.28; eight original studies) but not for positive symptoms (pooled SMD: −0.01, 95% CI: −0.12, 0.09; eight original studies), compared to the G allele [44] (Figure 4, Appendix A). Yoshikawa et al. found similar significant associations under the dominant model (i.e., CC versus G carriers) for both negative (pooled SMD: 0.56, 95% CI: 0.33, 0.80; five original studies) and positive symptoms (pooled SMD: 0.33, 95% CI: 0.05, 0.62; five original studies), while no additive models analyzed or raw data were presented [45] (Figure 4, Appendix A).

### 3.6. Antipsychotic-Induced Weight Gain

Overall, 8 out of 29 included studies performed meta-analyses on the association between 21 (*DRD2*, *HTR2A*, *HTR2C*, *HTR6*, *TNF-alpha*, *BDNF*, *ADRA2A*, *ADRB3*, *ANKK1*, *CNR1*, *FTO*, *GNB3*, *INSIG2*, *LEP*, *LEPR*, *MC4R*, *MDR1*, *MTHFR*, *PPARG*, *SNAP25*, and *CYP2D6*) genes and antipsychotic-induced weight gain [32,35,49,50,51,52,53]. The qualitative outcome was defined as ≥7% or ≥10% weight/BMI gain in these studies. One meta-analysis study (Ma, 2014 [49]) included both weight gain and metabolic syndrome. Around half of the studied genes (i.e., *DRD2*, *HTR2C*, *BDNF*, *ADRA2A*, *ADRB3*, *GNB3*, *INSIG2*, *LEP*, *MC4R*, and *SNAP25*) had at least one PGx variant showing significant association under at least one genetic model with weight gain. Another half (i.e., *HTR2A*, *HTR6*, *TNF-alpha*, *ANKK1*, *CNR1*, *FTO*, *LEPR*, *MDR1*, *MTHFR*, *PPARG*, and *CYP2D6*) did not exhibit any significant effect on antipsychotic-induced weight gain based on the meta-analyses results (Figure 3).

As shown by the network of these meta-analyses (Figure 2), the study by Zhang et al. (2016) was the largest by far, including the highest number of original studies (*n* = 72) on AIWG in their meta-analyses (Figure 2) [32].

#### 3.6.1. *DRD2*

Three PGx variants of *DRD2* (rs1799732*Del, rs6275*T, and rs7131056*A) were found to be significantly associated with AIWG under the additive model, with a pooled Hedge’s g of 0.31 (95% CI: 0.07, 0.54; three original studies), 0.25 (95% CI: 0.09, 0.41; four original studies), and 0.19 (95% CI: 0.03, 0.34; original studies), respectively (Figure 4, Appendix A). Similarly, they reported associations for PGx variant(s) of *BDNF*, *ADRA2A*, *ADRB3*, *GNB3*, *INSIG2*, *MC4R*, and *SNAP25* [32].

#### 3.6.2. *HTR2C*

Notably, five studies performed meta-analyses on *HTR2C* rs3813929 and AIWG (Sicard, 2010 [35], Ma, 2014 [49], Zhang, 2016 [32], Suetani, 2017 [53], Chen, 2020 [50]). Zhang’s meta-analysis study covered all original studies included in other meta-analysis studies except for Chen (2020), which included three more Chinese original studies and one original study published after Zhang’s meta-analysis study. However, the largest study (Zhang, 2016) only analyzed the recessive model comparing *HTR2C* rs3813929*CC to T carriers. They found a pooled Hedge’s g of 0.23 (95% CI: 0.04, 0.42; 20 original studies) and pooled OR of 1.96 (95% CI: 1.19, 3.22; 18 original studies). The raw data was not reported. Zhang et al. also showed that *HTR2C* rs518147*GG was associated with the risk of AIWG (compared to C carriers), with a pooled Hedge’s g of 0.18 (95% CI: 0.02, 0.34; five original studies) and a pooled OR of 1.86 (95% CI: 1.03, 3.35; five original studies) [32].

#### 3.6.3. *LEP*

Overall, three meta-analysis studies examined *LEP* rs7799039 (Shen, 2014 [51], Zhang, 2016 [32], and Yoshida, 2020 [52]) with AIWG. Yoshida and Shen identified significant associations in Asians, with the rs7799039*A allele being associated with an increased risk of AIWG. Shen reported a pooled OR of 1.58 (95% CI: 1.20, 2.07; four original studies) in Asians [51], and Yoshida reported a pooled OR of 2.32 (95% CI: 1.41, 3.82; two original studies) in Asians with the first episode of psychosis [44] (Figure 4, Appendix A).

### 3.7. Antipsychotic-Induced Metabolic Syndrome

Ma et al., 2014, examined three *HTR2C* PGx variants in relation to antipsychotic-induced metabolic syndrome, which was defined based on the definition of the National Cholesterol Education Program’s Adult Treatment Panel III or the International Diabetes Federation criteria. They found that *HTR2C* rs1414334*C carriers had a higher risk (compared to GG), with a pooled OR of 2.42 (95% CI: 1.48, 3.96; three original studies) [49]. Other genetic models were not analyzed (Figure 4, Appendix A).

### 3.8. Antipsychotic-Related Prolactin Level

Two meta-analysis studies looked into antipsychotic-related prolactin levels (Miura, 2016 [55] and Calafato, 2020) [56]. Miura et al. observed significantly increased levels of prolactin in *DRD2* rs1800497*A1 carriers (compared to A2A2) with a pooled Hedge’s g of 0.25 (95% CI: 0.07, 0.43; five original studies) [55]. Calafato et al. found no significant association between *CYP2D6* and antipsychotic-related prolactin level [56] (Figure 4, Appendix A).

### 3.9. Tardive Dyskinesia

In total, 4 out of 29 included meta-analysis studies investigated the association between four genes (*NQO1*, *SOD2*, *COMT*, and *BDNF*) and TD [57,58,59,60]. Two genetic variants (*COMT* rs4680 and *BDNF* rs6265) exhibited significant association with TD under a dominant or recessive model. Yet, no significant additive model effect was reported (Figure 4). The *COMT* rs4680*ValVal genotype was associated with an increased risk of TD only in women, compared to Met carriers (pooled OR: 1.63, 95% CI: 1.09, 2.45; seven original studies) [60]. Another meta-analysis study found that carrying the *BDNF* rs6265*Met polymorphism was associated with increased severity of TD (Abnormal Involuntary Movement Scale total score) in Caucasians compared to ValVal carriers (pooled Hedges’ g: 0.25, 95% CI: 0.03, 0.48; two original studies) or when comparing the MetMet genotype with Val carriers (pooled Hedges’ g: 0.58, 95% CI: −0.02, 1.14; two original studies) [57]. There was no significant association in other populations (Appendix A).

### 3.10. Clozapine-Induced Agranulocytosis

Islam et al. performed meta-analyses on 13 variants across nine genes (*NQO2, CYBA, MPO, HLA-DRB1, HLA-DQB1, HLA-DRB4, HLA-DRB5, HLA-DR2,* and *HLA-B*) and CIA, defined by ANC< 500/μL [61]. Only the association between *HLA-DRB1**04:02 and CIA remained significant after the application of the Bonferroni correction when carriers were compared to non-carriers (pooled OR: 5.89, 95% CI: 2.20, 15.80, two original studies) (Appendix A).

### 3.11. Pharmacokinetics of Antipsychotics

The metabolism of antipsychotics is regulated by CYP enzymes, especially *CYP2D6* and *CYP2C19*. There were three meta-analysis studies investigating the impact of CYP genes on the pharmacokinetics of antipsychotics (Takuathung, 2019 [64]: *CYP1A2* with clozapine and olanzapine; Zhang, 2019 [63]: *CYP2D6* with aripiprazole; Milosavljevic, 2021 [62]: *CYP2D6* with aripiprazole, clozapine, haloperidol, quetiapine, and risperidone, *CYP2C19* with clozapine). Takuathung et al. did not find any impact of *CYP1A2* PGx variants on the pharmacokinetics (concentration-to-dose ratio or concentration) of clozapine and olanzapine [64]. Zhang et al. reported that the dose-adjusted aripiprazole serum levels varied significantly between *CYP2D6* EMs and UMs and IMs but not between PMs and IMs [63]. Overall, the metabolism of aripiprazole is fastest in individuals who are UMs, followed by EMs and IMs, and is slowest in PMs. Similar findings were observed by another study (Milosavljevic, 2021 [62]), which showed a faster aripiprazole metabolizing rate in *CYP2D6* normal metabolizers (NMs, also called EMs) than IMs or PMs. The same trend was observed for risperidone [62]. Furthermore, *CYP2D6* PMs were also associated with higher haloperidol and quetiapine exposure (measured by dose-normalized steady-state plasma levels, dose-normalized area under the plasma level (time) curve, or apparent total clearance of the drug), compared to NMs. However, analysis of clozapine did not reveal any associations with *CYP2D6* but rather with *CYP2C19*, with PMs nearly doubling the mean value of clozapine exposure in NMs [62] (Figure 4, Appendix A).

## 4. Discussion

In the present study, we reviewed, meta-analyzed, and reported the evidence from meta-analysis studies on the associations between PGx variants and seven treatment outcomes of antipsychotics in patients with SSD. The most studied outcomes were treatment response and antipsychotic-induced weight gain, with the most studied genes were *DRD2* for response and *HTR2C* for AIWG. The most studied antipsychotic for response was clozapine, which is primarily prescribed to patients who do not benefit from other antipsychotics. As for AIWG, most meta-analysis studies included patients on various antipsychotics and analyzed them all together, which might limit the interpretation of the results.

The challenges surrounding the poor prognosis of patients with SSD persist. So far, the Royal Dutch Association for the Advancement of Pharmacy Pharmacogenetics Working Group (DPWG) has provided prescribing guidelines regarding six antipsychotics (aripiprazole, pimozide, risperidone, zuclopenthixol, brexpiprazole, and haloperidol) and *CYP2D6* phenotypes, of which only two (aripiprazole and brexpiprazole) have been deemed actionable by the US Food and Drug Administration (FDA) and the European Medicines Agency (EMA) [19]. Notably, the FDA’s recommendation to reduce clozapine dosage for individuals with poor *CYP2D6* metabolism was not supported by the meta-analyses [62], and the impact of *CYP2C19* phenotypes on clozapine metabolism is not reflected in the FDA’s drug label for this medication [62]. To date, other genes have not been considered in any guidelines due to the lack of convincing data. Unlike CYP genes, which are responsible for the pharmacokinetics of a drug and have a more straightforward impact on treatment outcomes, genes involved in the pharmacodynamics of antipsychotics play a more complex role in contributing to treatment outcomes. These genes often interact through multiple pathways, making it difficult and unreasonable to base recommendations solely on a single candidate gene.

It is well recognized that ethnicity can potentially impact the observed genetic associations in PGx studies; however, most of the studies in this field have a Eurocentric focus [66]. This may hinder the translation of research findings into clinical practice, especially in non-European populations. For instance, the effect of the *HTR2A* rs6313 allele on the response to clozapine was reported to be different between Caucasians and Asians [31,41]. The T allele was associated with better response in Caucasians, while in Asians, the trend was opposite but not statistically significant, with the C allele being the preferred allele. This highlights the potential differences in the impact of the same allele in different populations, which may be influenced by the variety of genetic backgrounds attributed to varying patterns of linkage disequilibrium and the discrepancies in allele frequencies. For example, *CYP* complex gene allele frequencies could vary greatly among ethnic groups, with *CYP2D6**10 existing in 58.7% of East Asians while only 0.2% of Europeans [67]. Furthermore, we could not rule out the influence of environmental and behavioral factors, as different food preferences and behaviors among ethnicities may influence the effect of a specific allele by affecting gene expression [68]. Furthermore, PGx variants can interact with each other and contribute to complex phenotypes. Therefore, the polygenic mechanism not only biases the results from studies analyzing mixed ethnic groups but also raises another concern about the clinical validity of candidate gene study results. Complex phenotypes such as clinical response and occurrence of side effects are multifactorial and influenced by both genetic and environmental factors, where genetic factors are further influenced by multiple alleles, each with a small to moderate effect size. Hence, the effect size reported for a single PGx variant in a candidate gene study may not be sufficient for clinical application.

In summary, conducting a well-designed PGx study requires considering, firstly, each antipsychotic is metabolized through not only a single gene but by several intertwined biological pathways. Each effector, including factors for gene regulation, RNA expression, post-translational modification, enzyme function, and protein structure, is regulated by sets of genes in interaction with each other and with the micro–macro environment. Secondly, beyond the gene–drug interaction, antipsychotics themselves interact with each other and with other concurrent medications through a labyrinthic relationship. It is reasonable to assume the interaction between drugs varies per each antipsychotic and the order of the medications. Furthermore, drug efficacy is highly affected by its consumed dosage, which is a direct outcome of patients’ adherence. It is conceivable that any misconduct, including ignoring, miscalculating, or underestimating any of these facts in PGx studies, will lead to highly biased results, deviated conclusions, and eventually less clinical applicability.

Genome-wide association studies (GWAS) are designed to provide an unbiased approach to the discovery of associations of a genome-wide set of variants with a trait, such as human characteristics, diseases, and drug response. For instance, a two-stage GWAS of up to 76,755 SSD patients and 243,649 control individuals has linked SSD to common variant associations at 287 distinct genomic loci [69]. However, limited attention has been given to therapeutic drug response and adverse drug reactions in genome-wide association studies, known as PGx GWAS [70]. This is due to the high sample size requirements and stringent threshold for statistical significance in GWAS, making it difficult to detect significant associations and identify alleles with small effect sizes.

Following the growth of GWAS, PRSs have gained popularity in predicting diseases and phenotypes, particularly after the success of PRSs in predicting the risk of SSD in 2009, which explained 3% of the variance [71]. PRSs also hold great potential for better prediction of treatment outcomes. A systematic review conducted by Johnson et al. in 2021 uncovered 51 papers investigating the use of PRSs for drug-related outcomes, and 30 of these focused on the treatment of psychiatric disorders [72]. However, most PGx PRSs are not derived from corresponding PGx GWAS but rather PRSs of the diseases of interest (e.g., SSD PRS for the prediction of response to antipsychotics) or non-drug-related phenotypes (e.g., obesity PGS for the prediction of AIWG). This may be because it is challenging to collect large patient cohorts receiving the same treatment. It is, therefore, unsurprising that inconsistent findings were reported, as the variants interacting with drugs were not adequately factored in [72]. Given the shortcomings inherent in the earlier strategies, future studies could refine PRSs by focusing on drug-related PGx variants or pathways with established evidence. It is reasonable to expect that a PRS constructed with sophisticated biological knowledge shall have a promising future, but well-designed articles are needed to confirm it. Furthermore, to produce reproducible results and facilitate the implementation of PGx-based personalized treatment, further considerations should be addressed, including the standardization of sources of DNA (saliva vs. blood), sample processing, and genotyping.

This study could serve as a starting point for constructing antipsychotic-related gene-based PRSs to predict treatment outcomes in patients with SSD. Additionally, this study also revealed the lack of meta-analyses on other important outcomes, such as cognitive function and specific metabolic outcomes (e.g., diabetes and high blood pressure). Despite limiting the scope of meta-analysis studies to those conducted after 2010, the study’s included meta-analyses have encompassed pre-existing research data, reducing the likelihood of missing important information.

## 5. Conclusions

In conclusion, while meta-analyses conducted over the past decade have identified several PGx variants of statistical significance for antipsychotic treatment outcomes in SSD, their small effect sizes and discrepancies across ethnicities suggest that their clinical relevance may be limited. Existing PGx guidelines are based on non-replicated findings from single studies, and the clinical application of PGx remains limited. To improve the accuracy and applicability of PGx in routine practice, further studies are necessary to integrate the strengths of PRSs with established knowledge and to consider the impact of polypharmacy on treatment outcomes. Overall, despite the growing volume of research, PGx has not yet reached its full potential in this area, and much work remains to be carried out to fully achieve personalized treatment for patients with SSD.

## Figures and Tables

**Figure 1 jpm-13-00471-f001:**
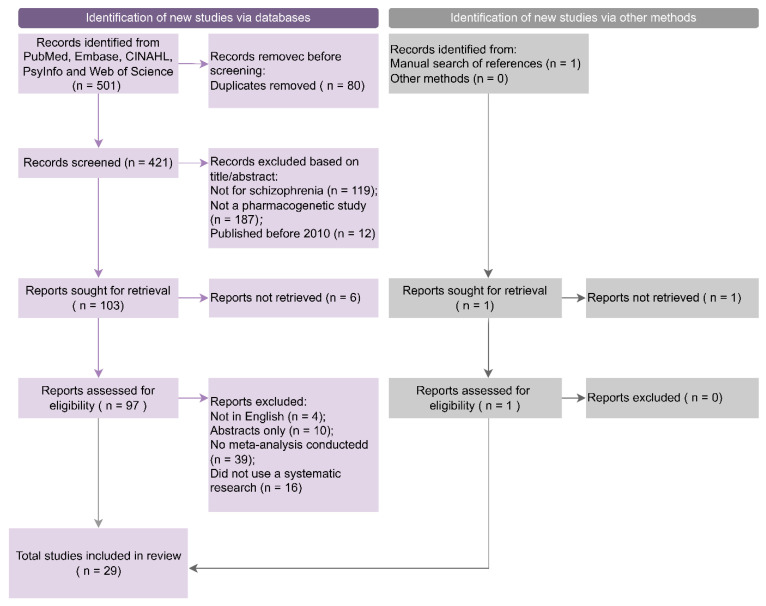
PRISMA flowchart of study selection and inclusion process.

**Figure 2 jpm-13-00471-f002:**
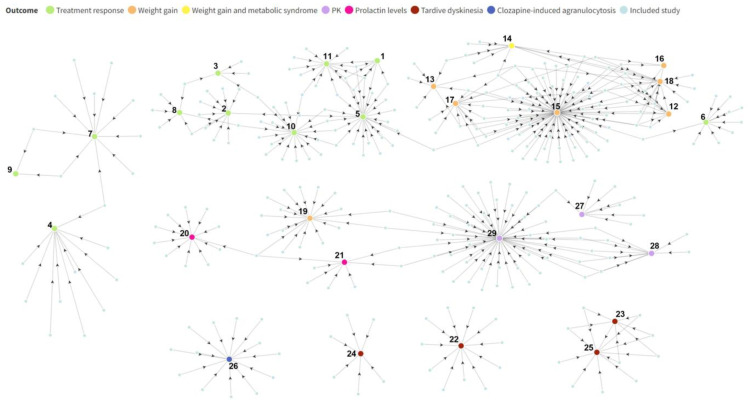
Network of the coverage of included meta-analyses. Footnote: Bigger dots represent meta-analyses included in this study; smaller dots represent original research studies included in each meta-analysis. Assorted colors indicate different outcomes. References: Study 1, Hwang, 2010 [34]; 2, Zhang, 2010 [46]; 3 de Matos, 2015 [47]; 4, Huang, 2016 [43]; 5, Gressier, 2016 [31]; 6, Cargnin, 2016 [40]; 7, Takekita, 2016 [44]; 8, Ma, 2019 [48]; 9, Yoshikawa, 2020 [45]; 10, Yan, 2022 [41]; 11, Liu, 2022 [42]; 12, Sicard, 2010 [35]; 13, Shen, 2014 [51]; 14, Ma, 2014 [49]; 15, Zhang, 2016 [32]; 16, Suetani, 2017 [53]; 17, Yoshida, 2020 [52]; 18, Chen, 2020 [50]; 19, Wannasuphoprasit, 2021 [54]; 20, Miura, 2016 [55]; 21, Calafato, 2020 [56]; 22, Zai, 2010a [59]; 23, Zai, 2010b [60]; 24, Miura, 2014 [57]; 25, Lv, 2016 [58]; 26, Islam, 2022 [61]; 27, Takuathung, 2019 [64]; 28, Zhang, 2019 [63]; 29, Milosavljevic, 2021 [62]. Abbreviation: PK, pharmacokinetic.

**Figure 3 jpm-13-00471-f003:**
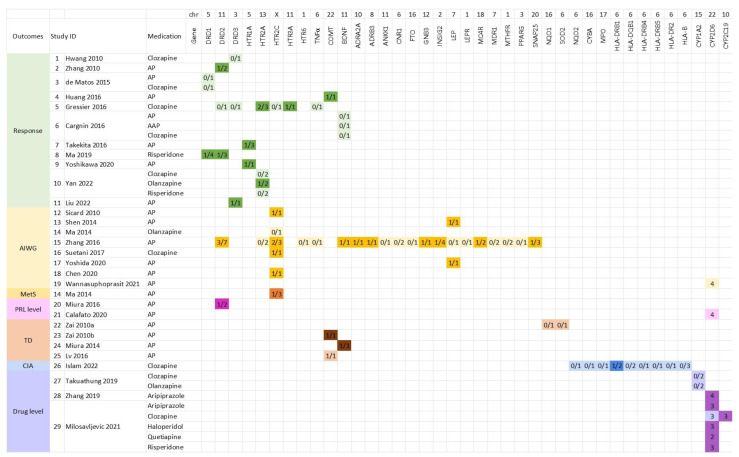
Heatmap of genes studied in each included meta-analysis. Footnote: Assorted colors indicate different outcomes. Lighter colors indicate that no SNP belonging to the gene was found to be significant for the corresponding outcomes/study. Darker colors indicate at least one SNP belonging to the gene exhibited a significant association with the corresponding outcome/Study. The numbers inside the boxes “a/b”: A presents the number of SNPs with significant associations; b presents the number of SNPs investigated in the specified study. For *CYP2D6* and *CYP2C19*, the number inside represents the number of phenotypes of this enzyme analyzed. References: Study 1, Hwang, 2010 [34]; 2, Zhang, 2010 [46]; 3 de Matos, 2015 [47]; 4, Huang, 2016 [43]; 5, Gressier, 2016 [31]; 6, Cargnin, 2016 [40]; 7, Takekita, 2016 [44]; 8, Ma, 2019 [48]; 9, Yoshikawa, 2020 [45]; 10, Yan, 2022 [41]; 11, Liu, 2022 [42]; 12, Sicard, 2010 [35]; 13, Shen, 2014 [51]; 14, Ma, 2014 [49]; 15, Zhang, 2016 [32]; 16, Suetani, 2017 [53]; 17, Yoshida, 2020 [52]; 18, Chen, 2020 [50]; 19, Wannasuphoprasit, 2021 [54]; 20, Miura, 2016 [55]; 21, Calafato, 2020 [56]; 22, Zai, 2010a [59]; 23, Zai, 2010b [60]; 24, Miura, 2014 [57]; 25, Lv, 2016 [58]; 26, Islam, 2022 [61]; 27, Takuathung, 2019 [64]; 28, Zhang, 2019 [63]; 29, Milosavljevic, 2021 [62]. Abbreviation: Chr, chromosome; AP, antipsychotics; AIWG, antipsychotic-induced weight gain; MetS, metabolic syndrome; PRL, prolactin; TD, tardive dyskinesia; CIA, clozapine-induced agranulocytosis.

**Figure 4 jpm-13-00471-f004:**
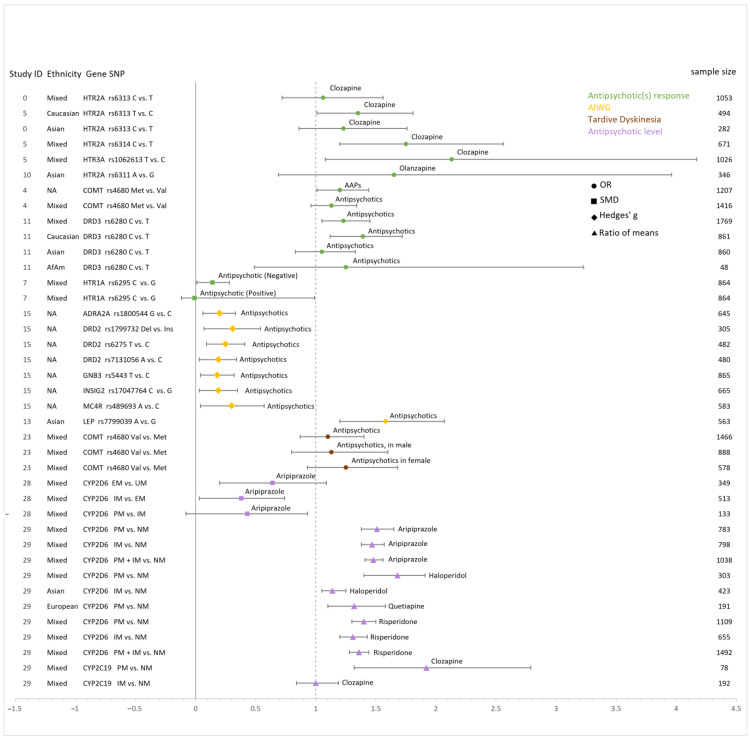
Pooled effect sizes of SNPs under the additive model. Footnote: Study ID is the same as the above figures. Study ID 0 represents the results recalculated in the present study by combining two or more meta-analyses. Each color of the marker in the middle of the error bar represents a different outcome. The shapes of markers mean different measurements of the effect size. Other significant SNPs without data under the additive model were not included in this figure. Abbreviation: AAPs, atypical antipsychotics; SMD, standardized mean difference; OR, odds ratio; AIWG, antipsychotic-induced weight gain.

## Data Availability

Not applicable.

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
