# Peer review of "The Progress and Pitfalls of Pharmacogenetics-Based Precision Medicine in Schizophrenia Spectrum Disorders: A Systematic Review and Meta-Analysis"

_jpm, 2023, doi:10.3390/jpm13030471_

Round 1
Reviewer 1 Report
The manuscript entitled: “The Progress and Pitfalls of Pharmacogenetics-Based Precision Medicine in SSD: A Systematic Review and Meta-Analysis” by Yuxin Teng et al., (jpm-2244460) describes the associations between pharmacogenomics (PGx) variants and treatment outcomes to antipsychotics in schizophrenia spectrum disorders (SSD).
This is an interesting and well-done comprehensive study. I have no questions for the authors.
Author Response
Dear Reviewer,
Thank you for taking the time to read and review my paper. I am delighted to hear that you found it to be well-done and that you have no further questions for me. Your positive feedback means a lot to me, and I appreciate the time and effort you put into reviewing my work.
Sincerely,
Yuxin
Reviewer 2 Report
A very interesting manuscript comparing the methods of personalized medicine in schizophrenia spectrum disorders. Interesting work, well collected material, nicely presented and well discussed.
I have a few minor remarks that I propose to implement to improve the quality of the publication:
- please remove the SSD from the title and use the full extension of the name.
- in the introduction, the authors discuss the methods of persanalyzed medicine in relation to SSD. Please provide a concise summary of these methods, in particular, please mention the classic monitored drugs with neurotropic effects, such as carbamazepine (there have already been published 20 years of registered studies).
- the value of each work is determined by clear conclusions. The manuscript will be read by thousands of medical professionals - specialists in clinical medicine and basic science, not just the authors. The conclusion must give a clear direction, tell about extensive genetic diagnosis before starting therapy, not think?, or maybe it cannot be clearly specified. When is it necessary to implement drug concentration-monitored therapy, and when it is not necessary. Of course, I do not expect the authors to answer all these questions in the conclusion, but I would like it to have a more practical aspect.
Author Response
Dear reviewer,
Thank you very much for your helpful comments and suggestions on our manuscript “The Progress and Pitfalls of Pharmacogenetics-Based Precision Medicine in SSD: A Systematic Review and Meta-Analysis”. We appreciate the time and effort you put into reviewing our work and providing us with valuable feedback.
We have carefully considered all of your comments and have made the requested revisions to the manuscript. In response to your comments, we have made the following changes:
- Checked and corrected some grammar and spelling errors in the article.
- Revised the title to include the full extension of the name instead of using the abbreviation SSD.
- Provided a summary of the current personalized treatment strategy in psychiatry, with specific examples.
- Revised the conclusion to provide clearer direction.
We believe that these revisions have improved the clarity and quality of our manuscript, and we hope that you will agree. We look forward to hearing your thoughts on the revised version of our manuscript.
Once again, thank you for your time and effort in reviewing our work.
Sincerely,
Yuxin